# Variation of Electrical Resistivity and Charge Passed in High-Performance Concrete

**DOI:** 10.3390/ma15196694

**Published:** 2022-09-27

**Authors:** Quang Tran, Pratanu Ghosh

**Affiliations:** 1Department of Pediatrics, Harvard Medical School, Boston Children’s Hospital, Brigham and Women’s Hospital, Boston, MA 02115, USA; 2Department of Civil and Environmental Engineering, California State University, Fullerton, CA 92831, USA

**Keywords:** pozzolan, durability, charge passed, resistivity, frequency, bridge decks

## Abstract

This study investigated the variation of bulk resistivity (*BR*) and charge passed for various high-performance concrete (HPC) mixtures based on significant factors (i.e., geometric size, operation frequency, and mixture constituents and proportions) using three testing instruments. These instruments were a surface resistivity (SR) meter and two bulk conductivity meters: one for using the *BR* data at a constant frequency, and the other at a wide range of frequencies. These HPC mixtures were categorized into several groups based on various supplementary cementitious materials (SCMs). The variation and distribution of *BR* and the charge passed were investigated and statistical analysis results showed that the addition of SCMs and their varying replacement level remarkably influenced the reduction of charge passed in each group over an extended period. The results revealed that, for fly ash-based ternary mixtures, the addition of 3% metakaolin or 12% silica fume resulted in the highest reduction of charge passed over time (82% and 90%, respectively). For cost purposes, 5% silica fume replacement in ternary mixtures was chosen as an optimal solution. Finally, this study offered promising options for charge passed computation to assess corrosion in light of simple, rapid, and reliable SR/BR measurement.

## 1. Introduction

Over the last few decades, several experimental techniques have been proposed to explore the electrical properties and durability of concrete structures due to chloride-induced corrosion [1,2,3]. These methods are relatively easy to perform due to their simple working procedure. One of the established destructive test methods currently performed based on electrical concepts is the Rapid Chloride Permeability Test (RCPT) following ASTM C1202 specification. During this procedure, the transport of chloride ions is accelerated by applying an external electrical field. The current passed through an electrical cell containing the concrete specimen is measured at five-minute intervals for 6 h. The resistance to chloride-ion penetration is then assessed by the total charge passed during the 6 h [4]. While this test is widely used, there are a few shortcomings that have been debated [5]. Nokken et al. [6] established electrical conductivity experimentation as a prequalification and quality assurance tool for concrete structures to measure durability performance. Their proposed testing methodology had several advantages over the traditional RCPT due to its short duration of testing in a non-destructive way, and it avoids the joule effect associated with the RCPT [6]. Bentz [7] developed a virtual rapid permeability test that replicates a real-world physical test. His research investigated a prototype virtual test method that included a prediction of the conductivity of the cementitious binder pore solution, and the total charge passed during an ASTM C 1202 RCPT [7]. Riding et al. [8] established a simplified method similar to the RCPT procedure to determine concrete resistivity as an index of concrete permeability. It should be noted that the specimen and gasket size were different compared to the RCPT. Only one current data point was taken (after 5 min) that could be applied to measure concrete resistivity to eliminate the gradual temperature rise problem. An empirical correlation between the new method and the standard RCPT method established the validity and promise of the new simplified method [8]. Recently, non-destructive testing, namely SR measurement, has become popular due to its characteristics of testing. An experimental investigation utilizing a Wenner Probe device on 529 sample sets was conducted by Kessler et al. to explore whether resistivity can be used as a quality control measure in place of the RCPT [9]. Rupnow et al. [10] recently showed the precision of a Wenner Probe resistivity meter during their experimental campaign of single laboratory and multi-laboratory measurements, and SR testing showed lower variability than rapid chloride permeability tests with various high-performance concrete (HPC) mixtures.

The chloride permeability of HPC concrete mixture is affected by the pore solution and the calcium silicate hydrate (C–S–H) content [11,12,13]. SCMs play an important role in chloride permeability by reducing pore volume and increasing CSH content by pozzolanic reactions. Shi et al. revealed that the use of SCMs results in a significant reduction in corrosion initiation [14]. Smith et al. [15] showed the significant advantage of using SCMs in the development of high-performance concrete. Their study exhibited that the use of SCM results in high values of concrete resistivity, which in turn reduces corrosion initiation [15]. Tikalsky et al. [16] conducted a recent study on different binary and ternary-based HPC mixtures’ electrical resistivity testing, and concluded that resistivity data correlated with RCPT data for various binary and ternary-based HPC mixtures. Marriaga et al. [17] studied the reliability of the RCPT and resistivity test based on the effectiveness of Ground Granulated Blast Furnace Slag (GGBFS) mixtures with various levels of cement replacements, and investigated their influence on the chloride resistance of concrete. It could be concluded from their study that the electrical resistivity and the total charge passed is an indirect measure of the chloride permeability suitable for both OPC and GGBFS mixtures. Recently, Ghosh et al. [18,19] conducted a study to demonstrate the effect of important factors on the surface and bulk resistivity of concrete, namely geometric size, probe spacing, and various levels of replacements of SCMs at different testing ages. As a result, a new recommendation was proposed for chloride ion permeability classification based on electrical resistivity data. However, most previous studies focus on a limited number of ternary mixtures for the computation of charge passed with a lack of variation of replacement of SCMs, and there is a clear gap in the current literature assessing the influence of significant parameters on the bulk electrical resistivity of concrete.

The use of the formation factor (FF) also addresses the impact of the pore solution resistivity on electrical measurements. This evolution permits the use of FF as the specification requirement without including the specified strength as the basis for selecting concrete mixtures for low chloride-ion penetrability [20]. FF has also been correlated with concrete sorptivity [21], and it can thereby be utilized to specify requirements for concrete mixtures that will be resistant to cycles of freezing and thawing from the perspective of the rate at which saturation may be achieved [22].

The comprehensive experimental campaign presented here focused on two aspects: (1) significant parameters (i.e., geometric size and operation frequency) influencing bulk resistivity, and (2) computation, reduction, variation, and distribution of the charge passed for fifty-one various ternary and binary-based high-performance concrete (HPC) mixtures in several groups using three different testing instruments. In addition, this study evaluated the influence of varying levels of replacement of silica fume and metakaolin on the reduction of charge passed in Class F and slag-based ternary concrete mixtures over an extended period. This research aimed to establish short-duration experimental and analysis techniques as a promising tool for durability investigations concerning chloride-induced corrosion and precision of the computation of charge passed data in an efficient way. The other purpose of this study was to identify multiple design solutions utilizing binary and ternary-based durable concrete mixtures that result in long-life bridge decks.

## 2. Materials and Methods

Fifty different types of ternary mixtures, including the binary and control mixture of 100% OPC with a water/cementitious materials ratio of 0.44, were designed to provide a wide range of testing data for this experimental program. The chosen water-cementitious material (w/c) ratio is typical of an exposed bridge deck and substructure concrete. Table 1 summarizes the constituents and proportions of all concrete mixtures. Limestone coarse aggregate of size 19 mm with a coarse aggregate factor (CAF) of 0.67 was chosen for all cementitious mixtures, and met ASTM C33 specification for gradation. In addition, ASTM C33 silica sand was used as a fine aggregate. Specific gravities of coarse aggregate and fine aggregate were 2.66 and 2.65, respectively. All SCMs were replaced by mass. The experiments were conducted using:Type II–V cement (TII-V) (moderate sulfate resistance cement of Type II and Type V blended) following ASTM C150 specificationGround granulated blast furnace slag of grade 100 (G100S)Ground granulated blast furnace slag of grade 120 (G120S)Class F fly Ash (F)Class C fly Ash (C)Silica fume (SF)Metakaolin (M)

California experiences major sulfate attack problems in concrete. Hence, we used Type II-V cement instead of Type I cement. The selection of different concrete mixture designs was based on the idea of satisfying basic technical properties and representing a diverse range of solutions for different durability problems.

The nomenclature of mixture parameters was chosen based on the percentage contribution by mass of each cementitious material, e.g., 75TII–V/20P/5SF means 75% Type II–V Cement, 20% volcanic pumice, and 5% silica fume.

A high-range water-reducer and an air-entraining admixture were utilized to satisfy better workability and durability performance specifications. Water reducer was used in the range of 10–14 oz per 100 lbs of cementitious materials, and air-entrainer was used in the range of 0.6 to 1.4 oz per 100 lbs of cementitious materials. All mixtures were cast according to ASTM C192 practice [23].

Three cylinders of 100 × 200 mm^2^ and two cylinders 150 × 300 mm^2^ were prepared for the bulk conductivity, SR measurement, and computation of the charge passed at the ages of 7, 14, 28, 56, and 91 days. The cylinders were demolded after 24 ± 2 h, and they were continuously cured in a lime water tank. A multiplier of 1.1 was used for the *BR* and SR data, as suggested by the AASHTO TP-95 specification for lime water curing conditions [24]. AASHTTO T358 was not available during the experimental investigation.

## 3. Charge Passed Models

### 3.1. Charge Passed from Bulk Conductivity Model following RCPT Theory

Typically, the charge passed obtained by the RCPT following ASTM C1202 [25] specification is the measure of chloride ion penetration in concrete structures. Recently, SR and *BR* testing methods have gained more attention due to their simple, reliable characteristics of testing over a very short period. The bulk conductivity and the charge passed measured by ASTM C1202 correlate, assuming the electrical current remains almost constant during the 6-hour test duration. The charge passed can still be computed within a few minutes following the RCPT theory utilizing the *BR* data (directly from *BR* measurement or converting the SR data to the *BR* data).

At the testing ages of 7, 14, 28, 56, and 91 days, 3 concrete cylinders were removed from the lime water tank and tested for SR and *BR* measurement. SR readings were obtained following the AASHTO TP-95 specification [25]. For each mixture, 24 SR data points (3 × 8 = 24 points) were obtained by 4-point Wenner Probe (refer to [18,26] for detailed measurement). *BR* readings measured by Merlin device (see Figure 1) were obtained twice by swapping two ends of the concrete cylinder within the clamp attached to the electrode plates and the data logger attached to the computer recorded the *BR*. Six conductivity data points (3 × 2 = 6 points) were collected by Merlin instrument (refer to [26] for detailed measurement). Throughout the testing time, the specimen ends were under fully saturated condition, except for the surface which was in wiped dry condition.

An alternating current source (325 Hz) was used to apply a current through the saturated cylinder. Similar to Merlin’s testing procedure, an RCON device (see Figure 1) also provided two readings by swapping the ends of the specimen quickly. Furthermore, the RCON measured the *BR* by application of a small-scale alternating current at different frequencies from 30 kHz to 1 Hz. Here, the frequencies applied were 10 kHz, 1 kHz, 100 Hz, and 10 Hz to observe a variation of the *BR* data over a wide range of frequencies. A total of 24 *BR* data points (4 × 2 × 3 = 24 points) at four frequencies of 10 kHz, 1 kHz, 100 Hz, and 10 Hz were collected by RCON tester. To accurately measure the bulk resistance, it was necessary to measure impedance over a range of frequencies, as the frequency is a function of hydration time and type of cement.

A voltmeter measured the voltage dropped across the specimen, and an ammeter measured the current through the specimen. From the measured current *I* and voltage *V*, the bulk conductivity was calculated as follows:(1)$$\sigma =\frac{I\times L}{V\times A}$$

The charge passed through a concrete specimen was computed from the electrical current using the following equation:(2)Q=I×t

From Equations (1) and (2), the charge passed was estimated from the *BR* or its inverse bulk conductivity from Equation (3):(3)$$Q_{M\left(\sigma \right)}=\sigma \frac{V\times A}{L}\times t$$
where $Q_{M\left(\sigma \right)}$ is the charge passed (*Coulomb*) obtained from bulk conductivity measurement by Merlin and RCON following the RCPT theory, *V* is the applied voltage (60 *V*), *A* is the area of the specimen (0.008 $m^{2}$), *L* is the length of the specimen (0.058 m), *t* is experimental testing time (6 h), and σ is bulk conductivity (${\mathrm{Sm}}^{-1}$). The bulk conductivity was derived directly from the measurement of the *SR* data with the application of a dimensionless geometric adjustment factor, expressed in Equation (4).
(4)$$Q_{M\left(SR\right)}=\frac{K}{\rho_{SR}}\times \frac{V\times t\times A}{L}$$
where $Q_{M\left(SR\right)}$ is the charge passed obtained from surface resistivity $\rho_{SR}$ (Kohm-cm), *K* is a dimensionless geometric correction factor (2.63) for the conversion of the *SR* to the *BR* [26,27]. Figure 1 shows the three different instruments.

### 3.2. Charge Passed Computation by Berke Model

Berke et al. [28] found that the electrical resistivity is related to the charge passed in concrete, and the relationship between them can be expressed by the following Equation (5) [28]:(5)$$\rho_{BR}=4887\times Q^{-0.832}$$
where Q is the charge passed, $\rho_{BR}$ is the bulk resistivity from Merlin tester, RCON meter, and Wenner Probe.

## 4. Results and Discussion

### 4.1. Influence of Geometric Size on the BR

To observe the impact of specimen size on the BR, nineteen out of fifty concrete mixtures were chosen for investigation. Every mixture included three small cylinders of 100 mm × 200 mm and two big cylinders of 150 mm × 300 mm for the BR measurement. Only two big cylinders were chosen for the BR measurement for each concrete mixture to avoid a large amount of material required for concrete mixing.

Figure 2 shows error bars that represent the variation of the ratios of the BR values at different frequencies at 7, 14, 28, 56, and 91 days. It was observed that the ratio was not dependent on time, and that variation in frequency from 10 kHz to 0.01 kHz had no impact on the BR.

This slight variation in the ratio (increment or decrement) was probably due to the setup of the instrument during the experimental investigation. As the RCON instrument uses the same bases or plates to test both the small and big cylinders, the diameter of the plates is bigger than that of the small cylinders. Therefore, there were some challenges in setting up the small cylinder for the BR measurement. Another possible reason was the variable wet conditions of the sponges. The sponge on the bottom surface was wetter compared to the sponge on the top surface. Occasionally, the sponge on the top surface was twisted, and some water dripped out of it as the upper base plate touched the surface of the cylinder due to the proper tightening procedure. The sponges needed to be squeezed as much as possible so that no water dripped on the surface of the cylinder during testing.

### 4.2. Influence of Frequency on the BR

Figure 3 and Figure 4 demonstrate the variation of BR at different frequency levels at 28 and 91 days. Figure 3 and Figure 4 show a definitive trend where the *BR* remains relatively constant at a frequency higher than 1 kHz, and it varies substantially in the frequency range of 0.01 kHz to 1 kHz. In addition, the results indicated that the effect of frequency becomes more significant for the *BR* greater than 30 Kohm-cm at a 1 kHz frequency level. In their previous study, Weiss et al. [3] stated that measurements at a fixed frequency may show measurable differences between various specimens; however, these differences are likely to include other features of the system than DC resistivity.

### 4.3. Charge Passed

Table 2 shows the average charge passed data of 50 mixtures at 91 days, which was computed using two different models as described in Section 3. The charge passed data were computed at 7, 14, 28, 56, and 91 days for long-term investigation. In Table 2, the first model derived from the RCPT theory is applied to Equation (3) and it considers the average bulk conductivity data by Merlin meter and the average bulk resistivity data at different frequencies by RCON tester. The first model is also applicable to Equation (4), which considers the average of SR data and the theoretical geometric correction factor K (2.63) to convert the SR to the bulk resistivity or its inverse bulk conductivity for computation of the charge passed [27]. The other empirical model developed by Berke et al. [28] applies to Equation (5), which uses the average BR by RCON tester and the average BR by Merlin meter, and Equation (4), which uses the average SR by Wenner Probe and geometric correction factor K (2.63) [26,28].

### 4.4. Charge Passed Reduction over an Extended Period

The charge passed reduces over time, and it is dependent on the type, the amount, and the pozzolanic reaction of SCMs in binary and ternary mixtures. In this study, the influence of the addition of Class C or Class F fly ash, silica fume, slag, and metakaolin with OPC on the reduction of charge passed was explored for binary mixtures. Furthermore, the effect of the second SCM on the charge passed reduction in ternary mixtures was also investigated. It is necessary to observe the charge passed reduction in the long-term investigation to understand the beneficial effect of various SCMs. The charge passed reduction is expressed in Equation (6).
(6)$$Reduction=\frac{Q_{t}-Q_{7}}{Q_{7}}$$
where $Q_{t}$ and $Q_{7}\;$ represent the charge passed at any time t (t > 7) and 7 days, respectively.

Figure 5, Figure 6 and Figure 7 express the charge passed reduction over time for ternary, binary, and OPC mixtures. Most mixtures had a common trend, where the reduction increased strongly until 28 days and slowly after 28 days. Furthermore, most binary and ternary mixtures achieved a 60% to 80% reduction in the charge passed at the age of 91 days.

Figure 5 presents the influence of Class C fly ash in the binary mixtures and the effect of 2nd SCM in Class C fly ash-based ternary mixtures. The binary mixture 80TII-V/20C had a higher reduction than the OPC mixture after 14 days. In addition, the combinations of Class C fly ash with class F fly ash or with silica fume provided a higher reduction than the binary mixtures from 14 days. However, only ternary mixtures of Class C fly ash with 35% G100S or with higher than 10% metakaolin replacement obtained better results than the binary mixture.

Figure 6 shows the influence of Class F fly ash in the binary mixtures and the effect of 2nd SCM in Class F fly ash-based ternary mixtures. The binary mixture 80TIIV/ 20F had a remarkably higher reduction compared to the OPC mixture after 14 days. Therefore, the addition of silica fume in ternary mixtures yielded more reduction than in binary mixtures over the long term. Some mixtures, namely 60TII-V/30F/10M and 45TII-V/15F/40G120S, had higher reduction till 28 days compared to the binary mixture.

Figure 7 indicates the influence of the second cementitious material on the charge passed reduction in silica fume-based ternary mixtures. The addition of fly ash in the SF-based mixtures yielded more reduction than the binary mixtures over time, except 65TII/5SF/30C. In silica fume-based ternary mixtures, the addition of 40% G120S obtained a lower reduction than the binary mixture at 91 days, whereas that of 35% replacement of G100S obtained a higher reduction up to 56 days than the binary ones.

### 4.5. Influence of the Variation of Metakaolin and SF Replacement on Charge Passed Reduction

Figure 8a,b depicts the effect of replacement levels of silica fume (3% to 12%) and metakaolin (3% to 12%) on the charge passed reduction in 20% Class F or 35% G120S-based ternary mixtures over 91 days. Figure 8a shows the combination of fly ash and 3% metakaolin obtained the largest reduction over time.

Any replacement level of silica fume in ternary mixtures provided a higher reduction than binary mixtures at all ages. As silica fume is costly, 5% silica fume was the optimum choice in Class F fly ash-based mixtures. Figure 8b shows that the addition of metakaolin or silica fume in 35% G120S-based ternary mixtures reduced more charge passed compared to the binary mixture, except for 10% of silica fume at 14 days. After 28 days, the combination of 35% slag G120S with replacement of 3%, 7%, and 12% metakaolin provided comparable results. For the combination of 35% G120S with silica fume, even though 12% silica fume provided the highest reduction at 91 days, overall, the reduction was comparable between 3% to 12% silica fume replacements.

### 4.6. Variation and Distribution of Charge Passed of Different Groups of Ternary Mixtures

In this study, three cylinders of each concrete mixture were measured for SR, BR, and charge passed computation. For each mixture, twenty-four SR data points (3 × 8 = 24 points) were obtained by 4–point Wenner Probe, six conductivity data points (3 × 2 = 6 points) by Merlin instrument, and twenty-four BR data points (4 × 2 × 3 = 24 points) by RCON tester at four frequencies of 10 kHz, 1 kHz, 100 Hz, and 10 Hz were collected. This study accumulated a wide range of charge passed data for fifty different concrete mixtures for statistical analysis to examine the variation of charge passed, utilizing a box plot and the distribution of charge passed utilizing frequency distribution in several groups. All the mixtures were categorized into four different groups based on the type of first SCM in ternary-based mixtures. All frequency plots excluded OPC as the purpose was to observe the distribution of charge passed for a particular group of the mixture.

The chloride ion permeability was classified into five classes based on the charge passed beyond 56 days or 3 months in some cases, according to the ASTM C1202 [25]. The classification is summarized in Table 3. For this reason, Figure 9, Figure 10, Figure 11 and Figure 12 show the variation in charge passed at 91 days. It was noticed that the vertical dashed lines in the box plots represent the limits of low and very low classes.

Figure 9a,b shows the variation and distribution of the charge passed of different Class C fly ash-based mixtures. From Figure 9a, the binary mixture is not significantly better than the OPC mixture. Most of the mixtures of silica fume and Class C fly ash obtained a very-low permeability class. Most of the mixtures of Class C with metakaolin or G120S had higher charge passed compared to the OPC mixture. Most Class C fly ash mixtures in combination with Class F fly ash performed better than those with metakaolin and G120S.

Figure 10a,b shows the variation and distribution of the charge passed of Class F fly ash-based ternary mixtures. All Class F fly ash mixtures with silica fume, metakaolin, or slag G100S obtained the charge passed in a very low permeability class. Overall, except for Class C fly ash, the addition of other SCMs helped Class F fly ash-based mixtures to obtain very low charge passed.

Figure 11a,b shows the variation and distribution of the charge passed in silica fume-based ternary mixtures. Most silica fume-based mixtures obtained the charge passed in the very low chloride ion permeability class. Most silica fume mixtures combined with slag or Class F fly ash had a lower charge passed than the mixtures of Class C fly ash and silica fume. The only outlier (60TII-V/5SF/35G120S mixture) had a low BR of 15.4 Kohm-cm at 91 days.

Figure 12a,b shows the variation and distribution of the charge passed of metakaolin-based mixtures in three different spectrums. It can be explained by the fact that the combinations of metakaolin with Class C mixtures had a wide range of charge-passed data (1110 to 2951 Coulombs). In general, the addition of Class C fly ash did not improve the quality of metakaolin-based ternary mixtures. The charges passed in most of these mixtures were higher than that of the OPC mixture, whereas Class F fly ash and G120S combined with metakaolin provided a significant beneficial effect. There were some outliers in the mixtures of metakaolin with Class F fly ash belonging to 60TII-V/10M/30F due to its low BR (13.1 Kohm-cm) at 91 days.

The reduction of charges passed observed in binary and ternary-based mixtures can be explained due to the chloride chemical binding effect [29] and the pozzolanic reactions [13]. Chloride ions can react with tricalcium aluminates (C_3_A) and C_4_AF to form calcium chloroaluminates and calcium chloroferrites, which are stable and lead to the reduction of available free chlorides [29]. The presence of fly ash (FA) increases the amount of C_3_A because of the increase of the alumina amount in the mixture [13]. Moreover, supplementary cementitious materials, such as fly ash, silica fume, metakaolin, and slag, can increase calcium silicate hydrate (C–S–H) content from the pozzolanic reactions. This increase in CSH content improves the binding capacity of the concrete matrix [30] and reduces the pore structure [14,18,31,32] and water permeability [33], leading to charge passed reduction.

## 5. Conclusions

The following conclusions were drawn according to the results of this study:The addition of SCMs in ternary mixtures, in general, helped to reduce the charge passed in a faster way compared to the binary mixtures. Specifically, in Class C fly ash-based mixtures, Class C fly ash combined with 35% slag G100S or with higher than 10% metakaolin obtained better results than the binary mixture. In metakaolin-based mixtures, the addition of 35% slag G120S are a better option for future bridge deck applications.In 20% fly ash-based ternary mixtures, the addition of 3% metakaolin or 12% silica fume resulted in the highest reduction of charge passed over time. 5% silica fume replacement in 20% fly ash in ternary mixtures can be considered an optimal solution. In addition, in 35% slag G120S-based ternary mixtures, the replacement of 12% silica fume provided the highest reduction at 91 days, and the addition of 3% to 12% silica fume provided a comparable reduction to each other. Furthermore, the addition of 3%, 7%, and 12% metakaolin provided a comparably higher amount of charge reduction over an extended period.The variation and distribution of charge passed data demonstrated that the addition of SCM provided a varying effect on the charge passed in the same group. Most ternary mixtures’ charge passed data fell within the range of low to very low chloride ion permeability class, except the ternary mixture of class C fly ash and metakaolin, which fell in moderate permeability.In practice, depending on the source and the availability of SCM, the first SCM in ternary mixtures design could be specified with confidence. Therefore, this study was significantly helpful in identifying the type and amount of second SCM in ternary mixtures to design low-charge-passed and high-durability concrete mixtures.The surface resistivity of concrete was determined by AASHTTO T 358, a newly developed standard, and the bulk resistivity of concrete was determined by AASHTTO TP 119 or ASTM C1876. All three instruments depicted consistency and accuracy in terms of computation of the charge passed for various concrete mixtures in an efficient way and within a very short period.The charge-passed distribution provided a useful tool for future research to compute the diffusion coefficients of HPC mixtures to assess chloride-induced corrosion.

## Figures and Tables

**Figure 1 materials-15-06694-f001:**
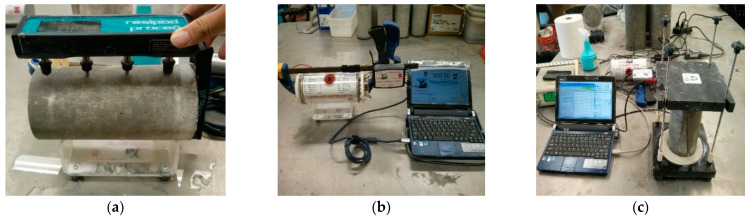
(**a**) 4 point Wenner probe (**b**) Merlin meter and (**c**) RCON tester.

**Figure 2 materials-15-06694-f002:**
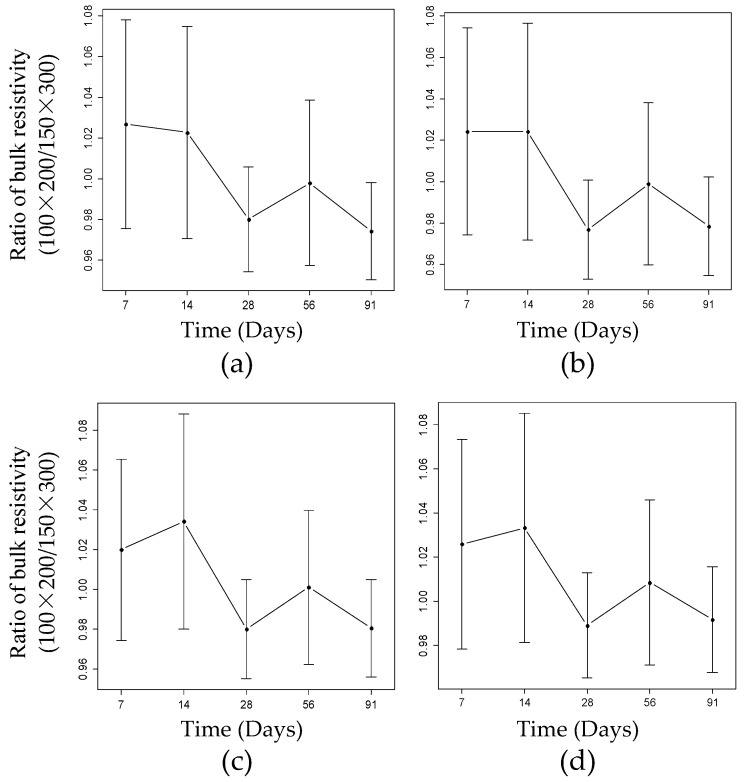
Influence of geometric size at (**a**) 10 kHz, (**b**) 1 kHz, (**c**) 0.1 kHz and (**d**) 0.01 kHz. The I-shaped line parallel to the y-axis is called an error bar. The point at the center of the line represents the mean of the ratios. The portion of the line from that point to the bottom and top represents the standard deviation of the ratios for nineteen mixtures.

**Figure 3 materials-15-06694-f003:**
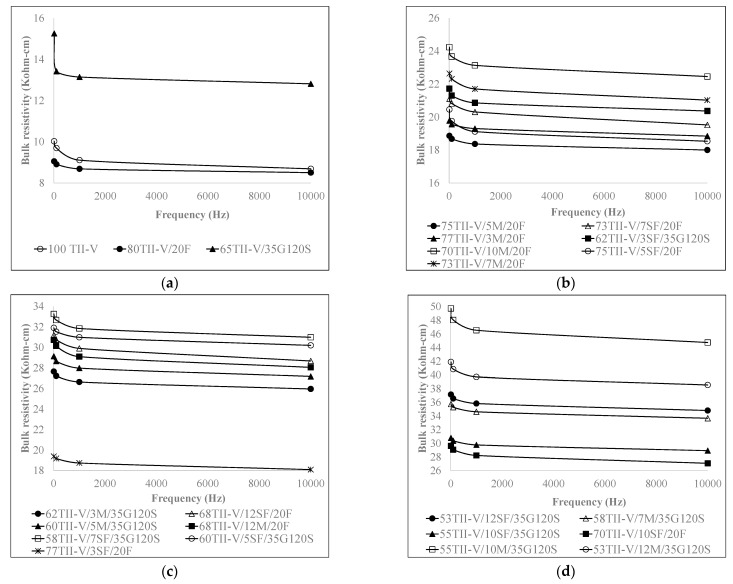
Influence of frequency on BR at 28 days on (**a**) control and binary mixtures, and (**b**–**d**) ternary mixtures.

**Figure 4 materials-15-06694-f004:**
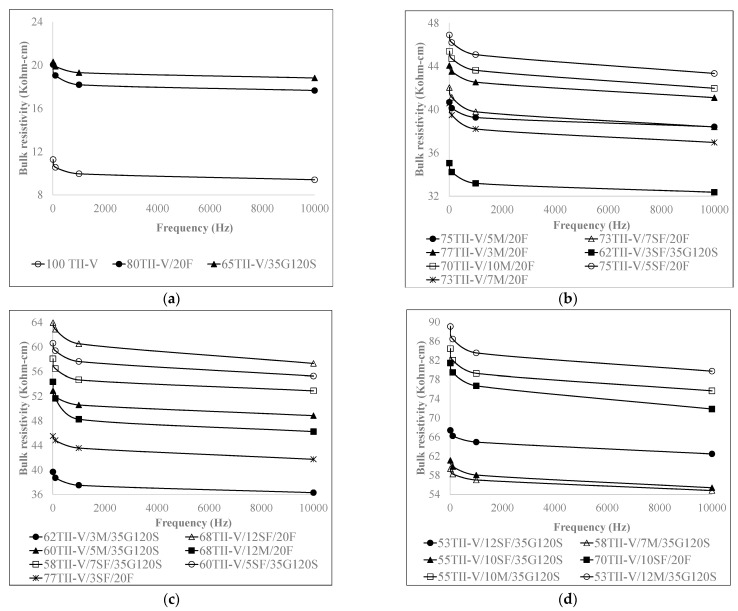
Influence of frequency on BR at 91 days on (**a**) control and binary mixtures, and (**b**–**d**) ternary mixtures.

**Figure 5 materials-15-06694-f005:**
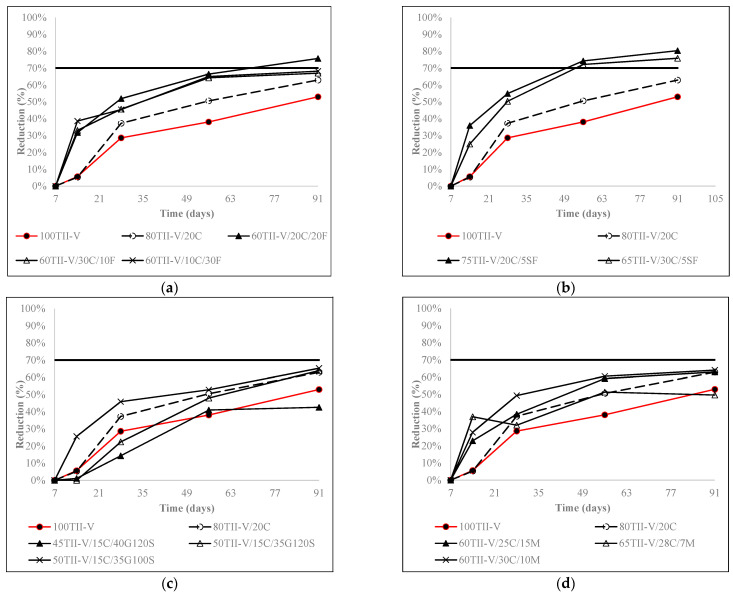
Influence of (**a**) Class F fly ash, (**b**) silica fume, (**c**) slag, and (**d**) metakaolin on the reduction of charge passed in Class C fly ash-based mixtures. The horizontal solid line represents the average reduction of charge passed for ternary mixtures at 91 days.

**Figure 6 materials-15-06694-f006:**
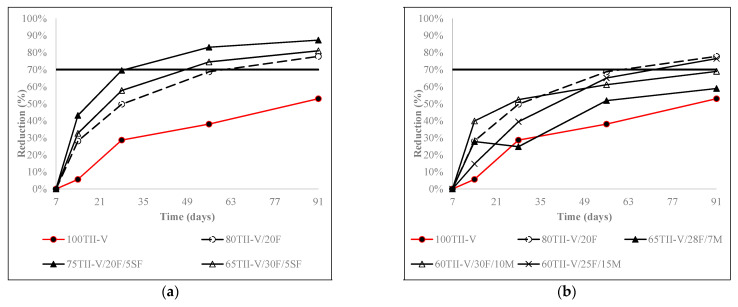
Influence of (**a**) silica fume, (**b**) metakaolin, and (**c**) slag on the reduction of charge passed in Class F-based ternary mixtures. The horizontal solid line represents the average reduction of charge passed for ternary mixtures at 91 days.

**Figure 7 materials-15-06694-f007:**
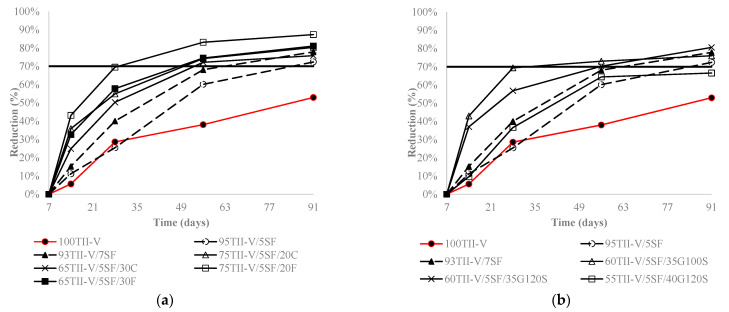
Influence of (**a**) fly ash and (**b**) slag on charge passed reduction in silica fume-based mixtures. The horizontal solid line represents the average reduction of charge passed for ternary mixtures at 91 days.

**Figure 8 materials-15-06694-f008:**
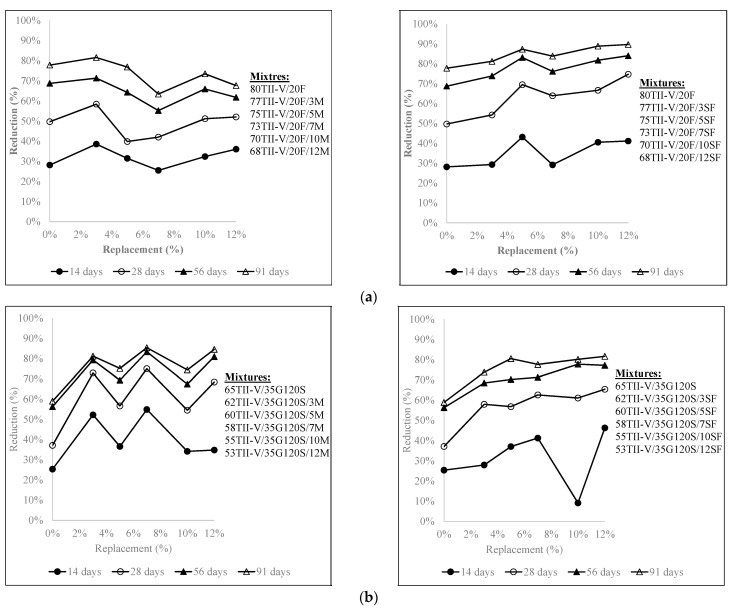
Influence of different replacements of silica fume and metakaolin in (**a**) 20% Class F and (**b**) 35% G120S mixtures.

**Figure 9 materials-15-06694-f009:**
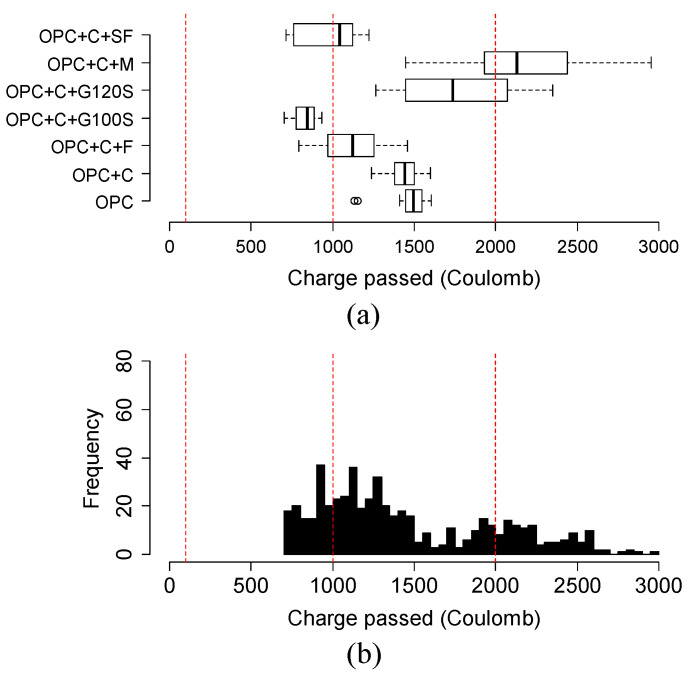
(**a**) Variation and (**b**) distribution of charge passed of Class C fly ash-based mixtures at 91 days. Three vertical red lines in the box plot represent the range of negligible, very low, and low permeability chloride ion permeability class as classified by ASTM C1202 specification.

**Figure 10 materials-15-06694-f010:**
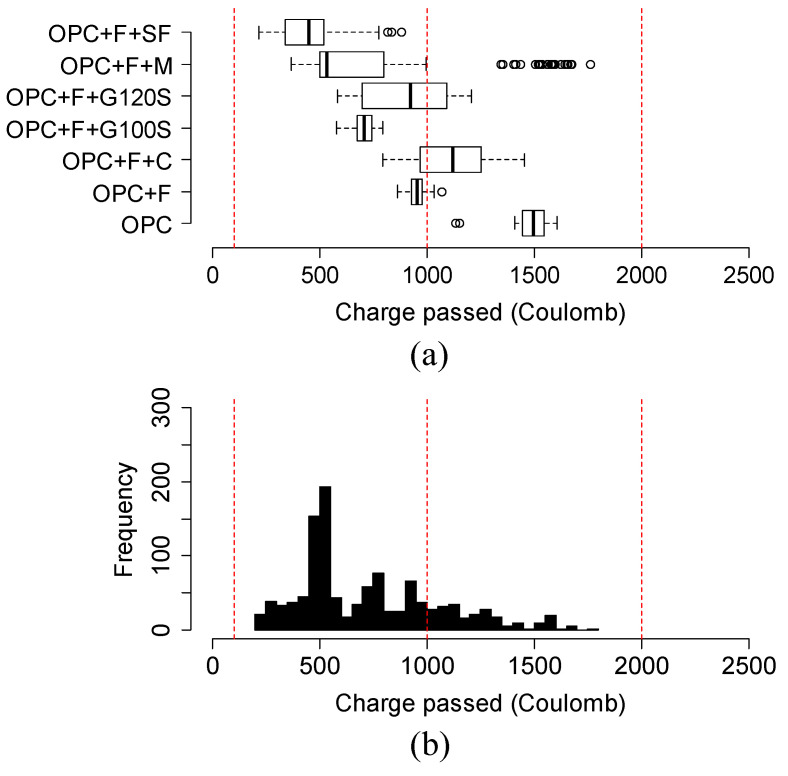
(**a**) Variation and (**b**) distribution of charge passed of Class F fly ash-based mixtures at 91 days. Three vertical red lines in the box plot represent the range of negligible, very low, and low permeability chloride ion permeability class as classified by ASTM C1202 specification.

**Figure 11 materials-15-06694-f011:**
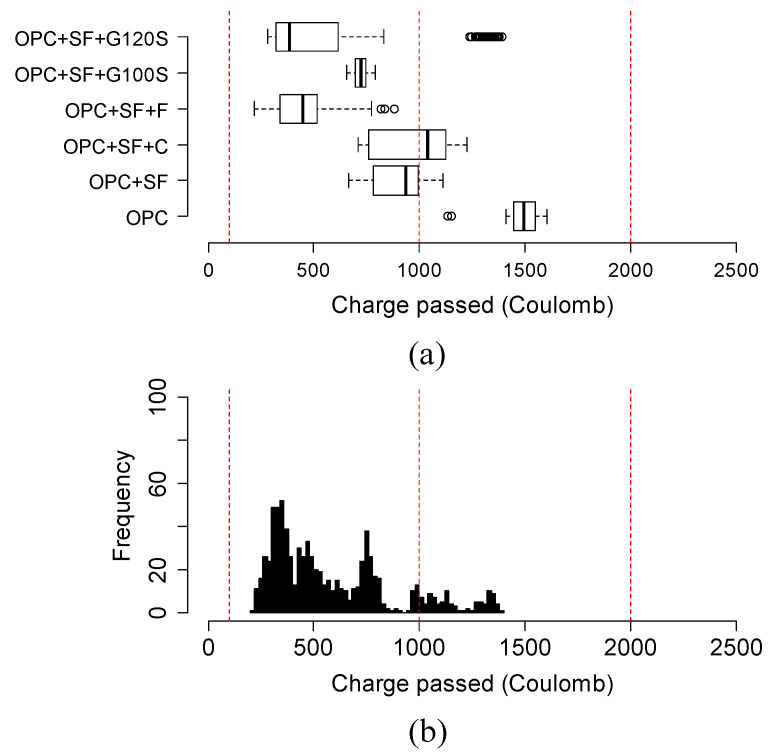
(**a**) Variation and (**b**) distribution of charge passed of silica fume-based mixtures at 91 days. Three vertical red lines in the box plot represent the range of negligible, very low, and low permeability chloride ion permeability class as classified by ASTM C1202 specification.

**Figure 12 materials-15-06694-f012:**
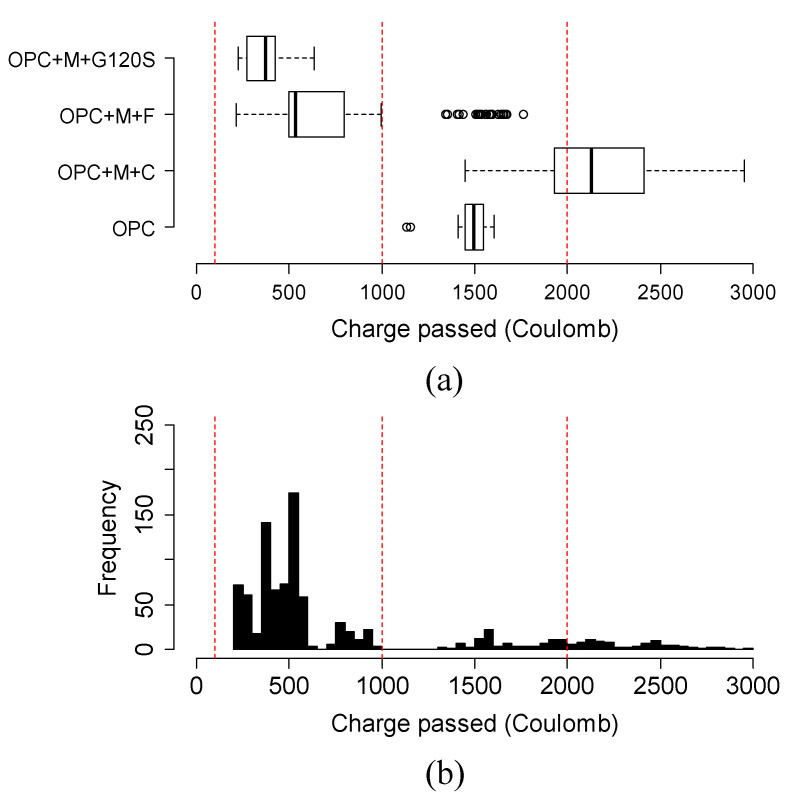
(**a**) Variation and (**b**) distribution of charge passed of metakaolin-based mixtures at 91 days. Three vertical red lines in the box plot represent the range of negligible, very low, and low permeability chloride ion permeability class as classified by ASTM C1202 specification.

**Table 1 materials-15-06694-t001:** Concrete mixture constituents and proportions.

No.	Mixture ID	28-Day Compressive Strength (psi)	Replacement (*%*)						
			Cement	Class C Fly Ash	Class F Fly Ash	Slag G100	Slag G120	Silica Fume	Metakaolin
1	100TII-V	3927	100	-	-	-	-	-	-
2	80TII-V/20C	3555	80	20	-	-	-	-	-
3	80TII-V/20F	4077	80	-	20	-	-	-	-
4	95TII-V/5SF	7841	95	-	-	-	-	5	-
5	93TII-V/7SF	7539	93	-	-	-	-	7	-
6	65TII-V/35G100S	5821	65	-	-	35	-	-	-
7	65TII-V/35G120S	4481	65	-	-	-	35	-	-
8	60TII-V/20C/20F	4230	60	20	20	-	-	-	-
9	75TII-V/20C/5SF	6974	75	20	-	-	-	5	-
10	60TII-V/25C/15M	5246	60	25	-	-	-	-	15
11	65TII-V/28C/7M	n/a	65	28	-	-	-	-	7
12	65TII-V/30C/5SF	7058	65	30	-	-	-	5	-
13	60TII-V/30C/10F	n/a	60	30	10	-	-	-	-
14	60TII-V/30C/10M	n/a	60	30	-	-	-	-	10
15	60TII-V/25F/15M	n/a	60	-	25	-	-	-	15
16	65TII-V/28F/7M	n/a	65	-	28	-	-	-	7
17	77TII-V/20F/3M	4341	77	-	20	-	-	-	3
18	75TII-V/20F/5M	4279	75	-	20	-	-	-	5
19	70TII-V/20F/10M	4196	70	-	20	-	-	-	10
20	75TII-V/20F/5SF	5264	75	-	20	-	-	5	-
21	70TII-V/20F/10SF	2432	70	-	20	-	-	10	-
22	73TII-V/20F/7M	6296	73	-	20	-	-	-	7
23	68TII-V/20F/12M	n/a	68	-	20	-	-	-	12
24	77TII-V/20F/3SF	n/a	77	-	20	-	-	3	-
25	73TII-V/20F/7SF	n/a	73	-	20	-	-	7	-
26	68TII-V/20F/12SF	n/a	68	-	20	-	-	12	-
27	60TII-V/30F/10C	4345	60	10	30	-	-	-	-
28	65TII-V/30F/5SF	5869	65	-	30	-	-	5	-
29	60TII-V/30F/10M	5276	60	-	30	-	-	-	10
30	60TII-V/35G100S/5SF	6318	60		-	35	-	5	-
31	50TII-V/35G100S/15C	6344	50	15	-	35	-	-	-
32	50TII-V/35G100S/15F	6138	50	-	15	35	-	-	-
33	45TII-V/35G100S/20F	4028	45	-	20	35	-	-	-
34	50TII-V/35G120S/15C	5629	50	15	-	-	35	-	-
35	50TII-V/35G120S/15F	4094	50	-	15	-	35	-	-
36	62TII-V/35G120S/3M	5498	62	-	-	-	35	-	3
37	60TII-V/35G120S/5M	6140	60	-	-	-	35	-	5
38	58TII-V/35G120S/7M	5594	58	-	-	-	35	-	7
39	55TII-V/35G120S/10M	5650	55	-	-	-	35	-	10
40	53TII-V/35G120S/12M	7263	53	-	-	-	35	-	12
41	50TII-V/35G120S/15M	n/a	50	-	-	-	35	-	15
42	62TII-V/35G120S/3SF	6742	62	-	-	-	35	3	-
43	60TII-V/35G120S/5SF	n/a	60	-	-	-	35	5	-
44	58TII-V/35G120S/7SF	4000	58	-	-	-	35	7	-
45	55TII-V/35G120S/10SF	4020	55	-	-	-	35	10	-
46	53TII-V/35G120S/12SF	5900	53	-	-	-	35	12	-
47	45TII-V/40G120S/15C	4600	45	15	-	-	40	-	-
48	45TII-V/40G120S/15F	5178	45	-	15	-	40	-	-
49	55TII-V/40G120S/5SF	5054	55	-	-	-	40	5	-
50	50TII-V/40G120S/10M	4695	50	-	-	-	40	-	10

Note: ‘n/a’ means missing data.

**Table 2 materials-15-06694-t002:** Charge passed by two models from three different instruments at 91 days.

No.	Mixture ID	Charge Passed following RCPT Theory (Coulombs)						Charge Passed by Berke Model (Coulombs)					
		From SR by Wenner Probe	From BR by Merlin	From BR (RCON)				From SR by Wenner Probe	From BR by Merlin	From BR (RCON)			
				10 KHz	1 KHz	0.1 KHz	0.01 KHz			10 KHz	1 KHz	0.1 KHz	0.01 KHz
1	100TII-V	1493	1487	-	-	-	-	1152	1148	-	-	-	-
2	80TII-V/20C	1454	1448	-	-	-	-	1116	1112	-	-	-	-
3	80TII-V/20F	926	960	-	-	-	-	649	678	-	-	-	-
4	60TII-V/20C/20F	1287	1287	-	-	-	-	964	965	-	-	-	-
5	60TII-V/30C/10F	1236	1256	-	-	-	-	918	938	-	-	-	-
6	60TII-V/30F/10C	1129	1132	-	-	-	-	823	828	-	-	-	-
7	75TII-V/20C/5SF	737	747	-	-	-	-	493	502	-	-	-	-
8	50TII-V/35G120S/15C	1472	1414	-	-	-	-	1133	1081	-	-	-	-
9	50TII-V/35G120S/15F	990	1097	-	-	-	-	703	796	-	-	-	-
10	95TII-V/5SF	948	991	-	-	-	-	667	705	-	-	-	-
11	93TII-V/7SF	922	786	-	-	-	-	645	534	-	-	-	-
12	65TII-V/5SF/30C	1258	1118	-	-	-	-	938	815	-	-	-	-
13	65TII-V/5SF/30F	864	771	-	-	-	-	597	521	-	-	-	-
14	55TII-V/5SF/40G120S	1241	1311	-	-	-	-	923	987	-	-	-	-
15	45TII-V/40G120S/15C	2154	2032	-	-	-	-	1789	1838	-	-	-	-
16	45TII-V/40G120S/15F	757	698	-	-	-	-	509	463	-	-	-	-
17	65TII-V/35G100S	961	962	-	-	-	-	679	681	-	-	-	-
18	60TII-V/35G100S/5SF	758	731	-	-	-	-	510	489	-	-	-	-
19	50TII-V/35G100S/15C	943	809	-	-	-	-	664	553	-	-	-	-
20	50TII-V/35G100S/15F	846	721	-	-	-	-	582	481	-	-	-	-
21	45TII-V/35G100S/20F	752	693	-	-	-	-	505	459	-	-	-	-
22	60TII-V/30F/10M	1611	1580	1442	1372	1288	1234	1262	1235	1239	1167	1082	1027
23	60TII-V/30C/10M	2815	2568	2301	2179	2009	1929	2469	2215	2173	2035	1845	1758
24	50TII-V/40G120S/10M	388	365	412	394	380	372	228	212	275	260	249	243
25	60TII-V/25F/15M	804	791	733	704	687	678	548	537	549	524	508	500
26	60TII-V/25C/15M	2429	2183	1940	1873	1836	1817	2067	1822	1770	1697	1657	1635
27	50TII-V/35G120S/15M	371	386	393	378	366	358	216	227	260	248	238	232
28	65TII-V/28F/7M	930	922	775	748	713	694	652	647	588	563	531	514
29	65TII-V/28C/7M	1933	1933	1582	1512	1422	1333	1572	1574	1385	1311	1218	1127
30	77TII-V/20F/3M	501	503	458	442	432	427	310	313	312	299	291	287
31	75TII-V/20F/5M	552	543	490	479	469	463	348	344	339	330	321	316
32	70TII-V/20F/10M	493	501	448	431	421	415	304	310	304	290	282	277
33	75TII-V/20F/5SF	503	466	434	417	407	401	312	284	293	279	271	266
34	70TII-V/20F/10SF	296	251	262	245	237	231	164	136	160	147	141	137
35	60TII-V/35G120S/5M	435	435	385	372	363	356	262	261	254	243	236	230
36	55TII-V/35G120S/10M	242	264	249	237	229	223	129	137	150	142	136	131
37	60TII-V/35G120S/5SF	299	301	340	326	317	310	167	169	219	208	200	196
38	55TII-V/35G120S/10SF	352	365	340	324	315	308	203	211	218	206	199	194
39	73TII-V/20F/7M	498	515	509	493	477	464	308	322	355	341	327	317
40	68TII-V/20F/12M	499	516	407	390	364	346	309	310	271	257	237	223
41	77TII-V/20F/3SF	519	494	451	432	420	414	323	304	306	291	281	276
42	73TII-V/20F/7SF	500	533	490	473	457	448	309	317	339	324	312	304
43	68TII-V/20F/12SF	341	346	328	311	299	294	195	205	209	196	187	183
44	62TII-V/35G120S/3M	547	570	519	502	486	474	344	363	363	348	335	326
45	58TII-V/35G120S/7M	358	364	344	330	323	317	207	211	221	211	205	201
46	53TII-V/35G120S/12M	254	248	236	225	218	211	137	134	141	133	128	123
47	62TII-V/35G120S/3SF	586	625	582	567	550	537	375	401	416	404	389	378
48	58TII-V/35G120S/7SF	422	379	356	344	333	324	252	190	230	221	213	206
49	53TII-V/35G120S/12SF	344	328	301	290	284	279	197	186	189	180	176	172
50	65TII-V/35G120S	1094	1110	1000	975	946	926	793	808	798	774	747	728

Note: ‘-’ means missing data as the RCON was purchased after two other instruments.

**Table 3 materials-15-06694-t003:** Chloride Ion permeability based on charge passed according to ASTM C1202 [25].

Charge Passed (Coulombs)	Chloride Ion Penetrability
>4000	High
2000–4000	Moderate
1000–2000	Low
100–1000	Very Low
<100	Negligible

## Data Availability

Not applicable.

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
