# Peer review of "Variation of Electrical Resistivity and Charge Passed in High-Performance Concrete"

_materials, 2022, doi:10.3390/ma15196694_

Round 1

Reviewer 1 Report

In this manuscript, the variation of bulk resistivity (BR) and charge passed for various high-performance concrete (HPC) mixtures with SCMs have been experimentally investigated based on geometric size, operation frequency, mixture constituents, and proportions by using three testing instruments. The paper has to be enhanced by applying the listed revisions:

1-      Line 59:  Add a reference number for “Smith et al.”.

2-      Line 69: Please give more information on the indirect measure of chloride permeability. GGBS can protect the rebar against chloride corrosion in seawater thanks to its chloride binding capacity. Can these apparatus measure this action correctly?

3-      Line 93: Why was such a low W/B ratio (0.44) chosen? In other words, why did the authors prefer to study HPC instead of normal or low strength concretes? These test methods are mostly needed for evaluating the existing, non-durable structures.

4-      28-day compressive strength can be presented in Table 1. This parameter is also important for evaluating the durability of the given mixtures.

5-      Under the Results and Discussion part, there are not enough references to compare the obtained measurements with the previous findings. All the measurements and results should be comparable with those of the previous studies. This can be needed major revisions in the manuscript.

6-      The lack of pore solution analysis can be evaluated as a weakness for the presented results. How the SCMs used in this study affects the pore solution? This can be better discussed in the manuscript by citing previous works.

7-      In Figs. 5, 6, and 7, the line located on the 70% level should be explained in the charts.

8-      Conclusions: Is there any advice for standards? This can further increase the impact of the study.

Author Response

Thanks for your time to review the manuscript. Please see the attachment.

Author Response

(The authors gave the same response as above.)

Reviewer 3 Report

Variation of electrical resistivity and charge passed in high-per-formance concrete

This study presents the variation of bulk resistivity (BR) and charge passed for various high-performance concrete (HPC) mixtures based on significant factors (i.e., geometric size, operation frequency, and mixture constituents and proportions) from three testing instruments.

Article is interesting. Few observations are given below;

Abstract need revision with some quantitative results.

Some more latest studies are required in the introduction section to further highlight the importance of this study.

Qian, Y., Yang, D., Xia, Y., Gao, H., & Ma, Z. (2021). Transport Properties and Resistance Improvement of Ultra-High Performance Concrete (UHPC) after Exposure to Elevated Temperatures. Buildings, 11(9), 416.

Smarzewski, P. (2020). Mechanical properties of ultra-high performance concrete with partial utilization of waste foundry sand. Buildings, 10(1), 11.

Liang, R., Huang, Y., & Xu, Z. (2022). Experimental and Analytical Investigation of Bond Behavior of Deformed Steel Bar and Ultra-High Performance Concrete. Buildings, 12(4), 460.

Table 1 how Concrete mixture constituents and proportions were designed with reference to the previous studies.

Section 4.1. Authors must state the increase or decrease with reference to some logics or the reasons.

List of abbreviations and symbols are missing.

Authors must summarized results in more systematic way with reference to the previous studies.

Author Response

(The authors gave the same response as above.)

Round 2

Reviewer 1 Report

It can be accepted.